# Archival bone marrow smears are useful in targeted next-generation sequencing for diagnosing myeloid neoplasms

**Daichi Sadato**[1,2☯], **Chizuko Hirama**[1,2☯], **Ai Kaiho-Soma**[1,2], **Ayaka Yamaguchi**[1], **Hiroko Kogure**[1], **Sonomi Takakuwa**[1], **Mina Ogawa**[1,2], **Noriko Doki**[3], **Kazuteru Ohashi**[3], **Hironori Harada**[3,4], **Keisuke Oboki**[2], **Yuka Harada**[1]*

**1** Clinical Research Center, Tokyo Metropolitan Cancer and Infectious Diseases Center, Komagome Hospital, Tokyo, Japan, **2** Center for Medical Research Cooperation, Tokyo Metropolitan Institute of Medical Science, Tokyo, Japan, **3** Division of Hematology, Tokyo Metropolitan Cancer and Infectious Diseases Center, Komagome Hospital, Tokyo, Japan, **4** Laboratory of Oncology, Tokyo University of Pharmacy and Life Sciences, Tokyo, Japan

☯ These authors contributed equally to this work.
* yharada@cick.jp

**Data Availability Statement:** Data cannot be shared publicly because of the privacy policy of Ethics Committee. Data are available from the

## Abstract

Gene abnormalities, including mutations and fusions, are important determinants in the molecular diagnosis of myeloid neoplasms. The use of bone marrow (BM) smears as a source of DNA and RNA for next-generation sequencing (NGS) enables molecular diagnosis to be done with small amounts of bone marrow and is especially useful for patients without stocked cells, DNA or RNA. The present study aimed to analyze the quality of DNA and RNA derived from smear samples and the utility of NGS for diagnosing myeloid neoplasms. Targeted DNA sequencing using paired BM cells and smears yielded sequencing data of adequate quality for variant calling. The detected variants were analyzed using the bioinformatics approach to detect mutations reliably and increase sensitivity. Noise deriving from variants with extremely low variant allele frequency (VAF) was detected in smear sample data and removed by filtering. Consequently, various driver gene mutations were detected across a wide range of allele frequencies in patients with myeloid neoplasms. Moreover, targeted RNA sequencing successfully detected fusion genes using smear-derived, very low-quality RNA, even in a patient with a normal karyotype. These findings demonstrated that smear samples can be used for clinical molecular diagnosis with adequate noise-reduction methods even if the DNA and RNA quality is inferior.

## Introduction

Gene mutations are essential prognostic factors in diagnosing and predicting the effect of therapy on myeloid neoplasms [1, 2]. Next-generation sequencing (NGS) is normally performed using genomic DNA from fresh or stocked frozen bone marrow (BM) cells (BMCs) [3, 4].

However, adequate quantities of BMCs cannot be obtained in some patients. In such cases, laboratory tests, including karyotyping and flow cytometry in particular, are prioritized;

ethics committee of Tokyo Metropolitan Komagome Hospital (contact via E-mail: rinri@cick.jp) for researchers who meet the criteria for access to confidential data.

**Funding:** This research was supported in part by Clinical Research Fund (R010302001) of Tokyo Metropolitan Government (https://www.metro.tokyo.lg.jp/), and JSPS KAKENHI (Grant Number JP20K07840) of Japan Society for the Promotion of Science (https://www.jsps.go.jp/). All grants were awarded to Y.H. The funders had no role in study design, data collection and analysis, decision to publish, or preparation of the manuscript.

**Competing interests:** The authors have declared that no competing interests exist.

therefore, gene abnormalities cannot be analyzed by NGS. However, BM smears have high priority for use in cytomorphological diagnosis, and because BM smear slides are stored after use, they are easily available, obviating the need for additional BMCs and DNA and RNA samples. While previous reports demonstrated that BM smear samples can used as a DNA source for PCR or Sanger sequencing, the quality of the results was not closely examined, especially with respect to their potential application to NGS. Using BM smears as a source of DNA and RNA for NGS would enable molecular diagnosis with small amounts of BM, even in patients without stocked cells, DNA or RNA. Previous studies examined the utility of slides containing biopsy samples as a source of DNA and RNA for target sequencing of lung adenocarcinoma [5] and thyroid cancer [6] and were able to provide profiles of gene mutations, including driver and drug-resistance mutations, suggesting that preserved or pretest samples can be used for NGS. However, in these cases, the samples were prepared using formalin-fixed, paraffin-embedded (FFPE) slides that allow preservation for extended periods of time unlike BM aspirate smears made by drying and alcohol-based fixation. Recently, target sequencing of genes associated with myeloid malignancies was tested using archived BM smears derived from a patient with acute myeloid leukemia (AML) [7]. While the analysis showed that smear slides for NGS can be used to create gene mutation profiles, it is still unclear whether they can provide insight into other myeloid malignancies, information about the deterioration of data, including gene-expression noise in smear samples, and the utility of RNA derived from this source. The present study analyzed the quality of DNA and RNA in BM smear samples and assessed their utility in NGS analysis by analyzing the character of the variants detected.

## Materials and methods

### Ethics statement

All the procedures performed in the present study involving human participants were approved by the ethics committee of Tokyo Metropolitan Komagome Hospital, and all the patients provided written informed consent for participation.

### Patients and BM samples

Smear slides were prepared from diagnostic BM aspirates from which mononuclear cells were isolated and were stored at room temperature in a dark place. Genomic DNA from the mononuclear cells was extracted using Gentra Puregene Blood Kit (Qiagen, Hilden, Germany) in accordance with the manufacturer's instructions. Cells on the smears were harvested by scraping and using ATL buffer (Qiagen), and the DNA was purified using QIAamp DNA Mini Kit (Qiagen) in accordance with the manufacturer's instructions. RNA was extracted using TRIzol RNA Isolation Reagents (Thermo Fisher Scientific, Waltham, MA, USA). The integrity of the extracted RNA was determined using the 2100 Bioanalyzer (Agilent Technologies, Santa Clara, CA, USA), and the RNA integrity number (RIN), an algorithm for assigning an integrity value to RNA [8], was calculated using 4150 TapeStation (Agilent Technologies). The RNase P gene copy number in the genomic DNA was measured using TaqMan RNase P Detection Reagents Kit (Thermo Fisher Scientific) in accordance with the manufacturer's instructions.

### Targeted sequencing

Targeted sequencing was performed using AmpliSeq for Illumina Myeloid Panel (Illumina, San Diego, CA, USA) and a custom-designed panel to detect mutations in 68 genes and fusions of 29 driver genes (S1 Table). As a template, 10 ng DNA (for mutations) or cDNA synthesized from 10 ng RNA (for fusions) was used to amplify the target genes. AmpliSeq Library

Plus for Illumina (Illumina) was used to generate libraries. The size of the fragment libraries was determined using the 2100 Bioanalyzer. The libraries were analyzed using the MiniSeq High Output Reagent Kit (300 -cycles) with the MiniSeq (Illumina) platform in accordance with the manufacturer's instructions.

### Detection of variants and fusion genes

FASTQ files were generated, then cleaned with Trimmomatic [9], and the results were aligned to the human reference genome, hg19, using Burrows-Wheeler Alignment (BWA) [10]. Mapped reads and their coverages were analyzed using Qualimap [11]. Gene variants were detected using HaplotypeCaller (for high frequency variants) and Mutect2 (for low frequency variants) included in GATK [12]. Gene variants obtained from HaplotypeCaller were filtered with the parameters of quality/depth, mapping quality, and strand bias to exclude false-positive variants as previously described [13]. Variants were detected using the tumor-only mode or the panel of normal mode on Mutect2. The variants detected by Mutect2 were filtered with GATK FilterMutectCalls. ITDseek [14] and Pindel [15] were used to detect FLT3-ITD mutations. Variants were annotated with information from the Refseq, 1000G and Exac databases in Illumina VariantStudio 3.0 software (Illumina). Variants with a prevalence greater than 1% in a given regional population were excluded. Finally, mutations in hematological malignancies were manually analyzed. The FASTQ files cleaned with Trimmomatic were analyzed with JAFFA [16] and STAR-Fusion with FusionInspector [17] to detect fusion genes.

### Statistical analysis

A two-group comparison of DNA concentration value, RNase P copy numbers, and coverage analysis data was done using the Mann-Whiney U test with R (The R Foundation for Statistical Computing, Vienna, Austria) and GraphPad Prism (Graph Pad Software, CA, USA).

## Results and discussion

### Smears served as DNA sources for targeted DNA sequencing

Five paired samples of BMCs and BM smears were compared in terms of the quality of extracted DNA (Table 1).

The dsDNA/total DNA ratio in each sample indicating the degree of DNA decay was significantly lower (P = 0.0079) in the smear samples than in the BMCs (Fig 1A). At the same time, the copy number of the RNase P gene in 1 ng DNA was also significantly lower (P = 0.0079) in the smear samples (Fig 1B).

Although the DNA quality was lower in the smears than in the BMCs, it was sufficient to generate NGS libraries (S1A, S1B Fig). The libraries were analyzed, and the reads were mapped

**Table 1. Paired samples of bone marrow cells and smears.**

| Patient | Clinical diagnosis | Smear sample | BMC sample | Elapsed years |
|---------|--------------------|--------------|------------|---------------|
| #106 | MDS | Unstained | Frozen | 4 |
| #113 | MDS | MGG-stained | Frozen | 11 |
| #181 | MDS suspected | Unstained | Fresh | Fresh |
| #184 | AA | Unstained | Fresh | Fresh |
| #220 | t-AML | Unstained | Frozen | 0.1 |

MDS, myelodysplastic syndrome; AA, aplastic anemia; t, therapy-related; AML, acute myeloid leukemia; MGG, May-Grünwald Giemsa.

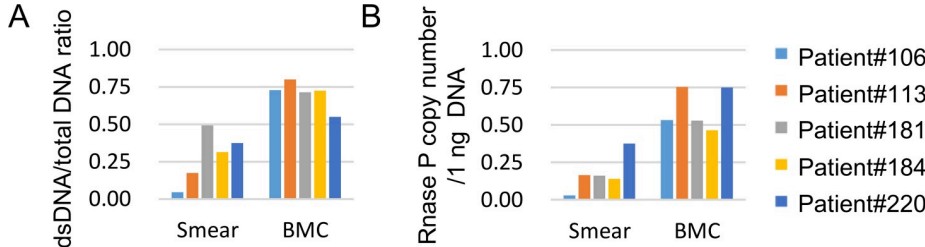

**Fig 1. Quality of smear- and Bone Marrow Cell (BMC)-derived DNA samples.** (A) dsDNA/total DNA ratio of the smears and BMC samples. (B) Copy number of the RNase P gene detected in smear and BMC samples.

to a human reference genome to evaluate the quality of the smear-derived sequence data. There was no difference between the smears and BMCs in terms of the total reads of the BAM file (Fig 2A, P = 0.2220), coverage (Fig 2B, P = 1.0000), and uniformities (Fig 2C, P = 0.8571). Furthermore, each amplicon was equally covered with synthesized reads (Fig 2D). These results suggested that the libraries of targeted sequences synthesized from smear-derived DNA are comparable with those synthesized from BMC-derived DNA.

Next, using these mapped sequences, variants were detected in paired samples using HaplotypeCaller (for germline or large clone variants) and Mutect2 (for somatic variants). Over 95% of variants detected via HaplotypeCaller were shared variants, while smear- and BMC-unique variants (3.08%) were suspected of being sequencing errors (Fig 3A). To investigate the characteristics of the variants, they were plotted according to their variant allele frequency (VAF) and read depth. Smear- and BMC-unique variants exhibited low read depth (Fig 3B). After these

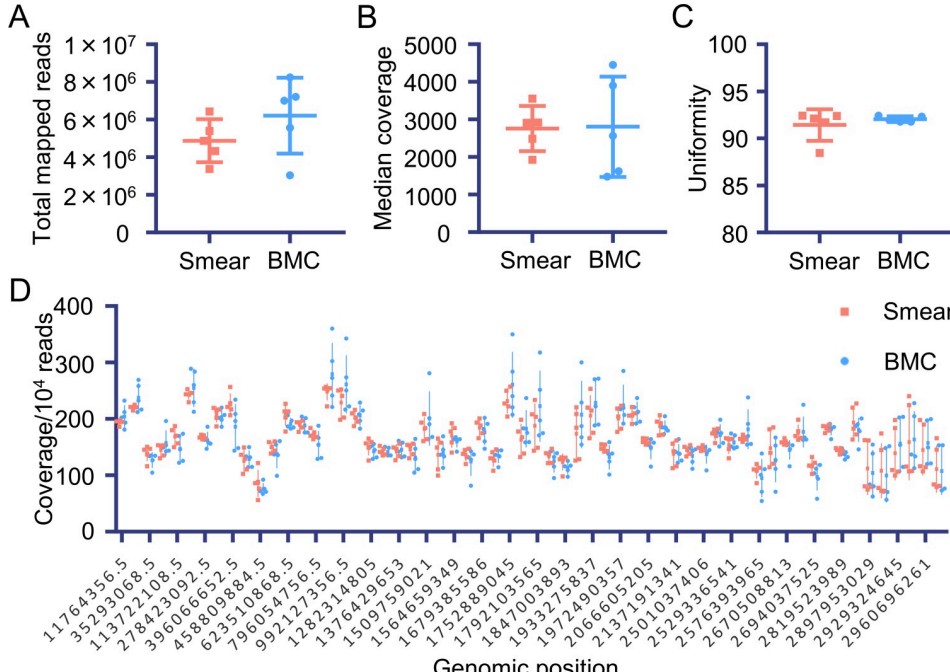

**Fig 2. Read and coverage analysis of target sequence data derived from the smear and BMC samples.** The following values were compared between the smear and BMC samples: (A) total mapped reads, (B) median coverage depth, (C) uniformity (more than 20% median coverage), and (D) normalized coverage for each amplicon region.

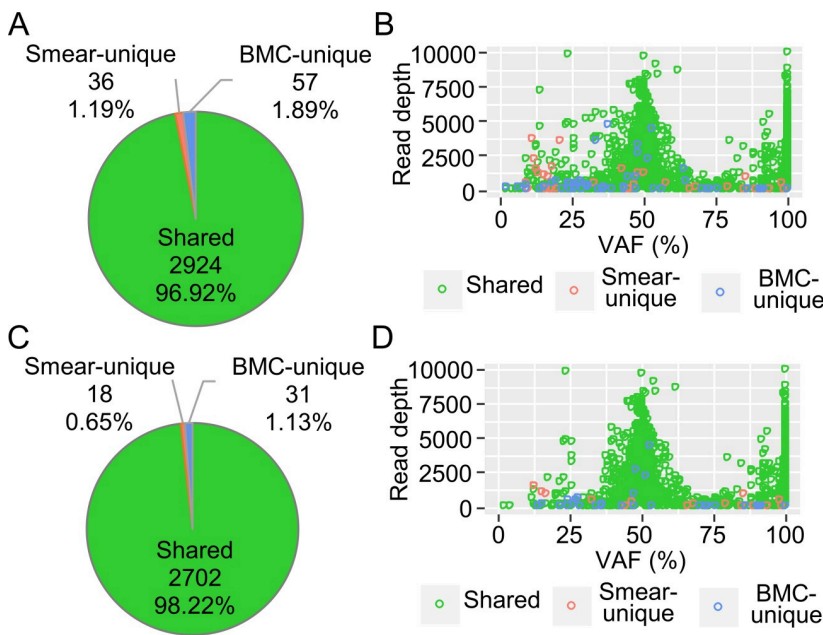

**Fig 3. Characteristics of variants detected using HaplotypeCaller and the filtering effect.** The combined results from five paired samples of smears and bone marrow cells (BMCs) are shown. (A) Pie chart of the smear-unique variants, BMC-unique variants, and shared variants. (B) Distribution of the detected variants. VAF, variant allele frequency. The smear-unique, BMC-unique, and shared variants are color-coded. (C) Pie chart of the variants after filtering. (D) Distribution of the filtered variants.

variants were filtered out, these variants decreased to 1.78%, and high VAF mutations were successfully detected in both the smear and BMC samples (Fig 3C and 3D).

However, the smear-unique variants detected by Mutect2 comprised two-thirds of the whole and needed to be filtered out (Fig 4A and 4B). The distributions of the large VAF variants showed two peaks at 100% and 50% VAF comprising chiefly SNPs while variants with a VAF of 25% or lower mostly consisted of small clusters of chiefly somatic variants. Smear-unique variants appeared to accumulate in very low-VAF regions, suggesting that they were noise (Fig 4B). FilterMutectCalls filtering was able to reduce this noise mainly by excluding low read depth noise; however, much smear-unique noise with a low VAF remained (Fig 4C and 4D).

To remove the artificial noises and detect variants sensitively using Mutect2, a panel of normals (PON) is recommended [18]. To apply this method in the present study, a PON was constructed by merging 13 BMC-samples from patients without myeloid malignancies, and the detected variants were plotted based on their VAF and read depth (Fig 5A). Variants were color-coded to indicate whether or not they were a SNP. Most variants with a suspected SNP accumulated at the 50% and 100% VAF peaks whereas the others were distributed mainly in the low-VAF regions. Most, though not all, of the noise was removed by FilterMutectCalls, suggesting that the remaining noise may have been an artifact of the assay (Fig 5B). Using the PON, the remaining noise was removed by subtraction, which effectively reduced the noise where the VAF was around 10%. However, smear-specific noise remained in areas with VAF <5% (Fig 5C). Since the smear-derived mutations accumulated in the low-VAF regions, VAF filtering was considered effective. To set the VAF threshold for eliminating noise, the VAF distributions of the variants left after subtraction were plotted (Fig 5D). Large amounts of the smear-unique variants accumulated in the low-VAF regions, especially where the VAF <2.5%, suggesting that this value can be used as the threshold value (Fig 5E).

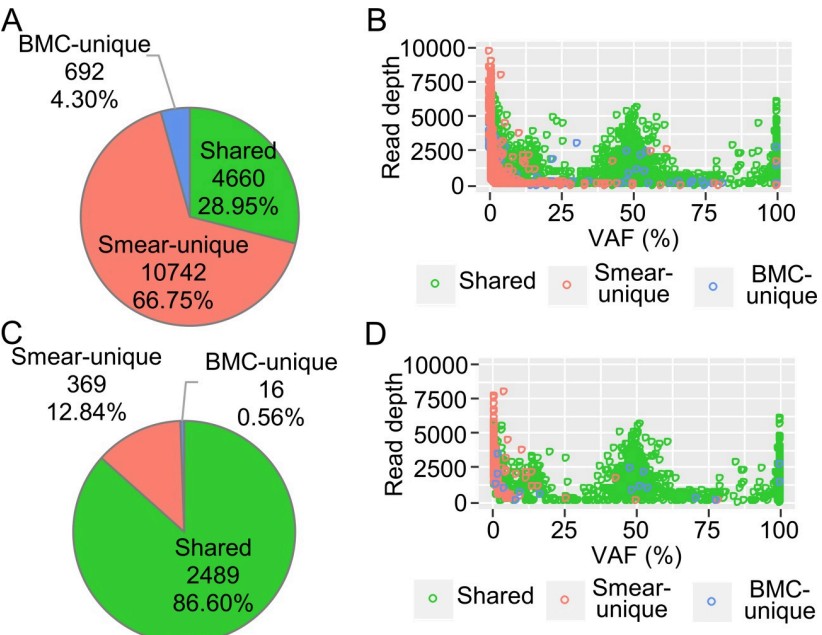

**Fig 4. Characteristics of the variants detected using Mutect2 and the filtering effect.** Combined results from five paired samples of smears and bone marrow cells (BMCs) are shown. (A) Pie chart of the smear-unique, BMC-unique, and shared variants. (B) Distribution of the detected variants. VAF, variant allele frequency. The smear-unique, BMC-unique, and shared variants are color-coded. (C) Pie chart of the variants after filtering with FilterMutectCalls. (D) Distribution of the filtered variants.

PON subtraction and VAF filtering, in addition to FilterMutectCalls filtering, effectively reduced the rate of smear- and BMC-unique variants (Fig 6A) and improved the distribution of the remaining variants (Fig 6B). Furthermore, the shared variants showed almost the same VAF values for the smear and BMC samples (Fig 6C).

These results suggested that BM smears can be used for targeted DNA sequencing even if they are stored at room temperature under normal laboratory conditions. In variant detection, a very little noise was found while using HaplotypeCaller, which is able to detect germline mutations and large clone size mutations accurately without extra filtering (Fig 3). On the other hand, Mutect2, which has high sensitivity for low VAF variants (e.g., somatic mutations), required modified filtering because many noises with low VAF, which were unable to be removed completely by default filtering, were detected in the smear samples (Fig 4).

Based on our results, we performed additional targeted DNA sequencing using smear samples from patients with myeloid neoplasms, mainly acute myeloid leukemia (AML) and myelodysplastic syndromes (MDS). Twenty-one samples preserved for 0.1–11 years were analyzed using the established method described above, then filtered (Fig 7A and 7B). Of the filtered variants, 8.53% were in exons or splice sites and had various VAFs (Fig 7C and 7D). The effect of the duration between the sample preparation stage and the assessment of DNA quality and variants was further analyzed to determine the utility of the archival smears. The quality of the extracted DNA was clearly unaffected by either the duration (Fig 8A) or staining (Fig 8B, P = 0.2773), suggesting that DNA can be extracted from various types of smear. However, regarding the results based on old smear samples, filtering for variants using either FilterMutectCalls or a 2.5% or lower VAF detection level showed a tendency towards increasing variants. On the other hand, no significant difference was found in the quantity of variants after filtering (Fig 8C). To identify the effect of staining smear samples on variant calling results,

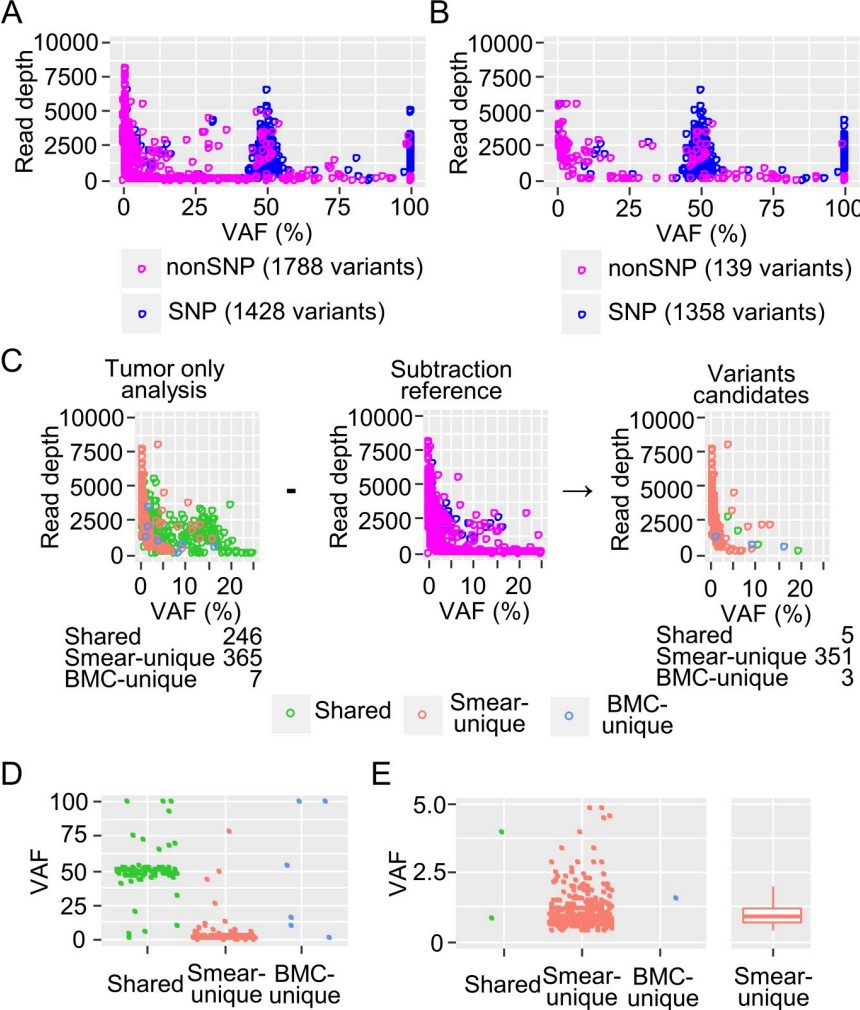

**Fig 5. Panel of Normals (PON) subtraction method and evaluation.** Distribution of all the variants (A) and the filtered variants (B) detected by PON. Variants with SNP and the other variants were color-coded. (C) Distribution of the subtracted variants with a low VAF. (D) VAF plot of shared, smear-unique, and BMC-unique variants after subtraction. (E) VAF plot of subtracted variants with VAF <5%. A boxplot of smear-unique variants is also shown.

detected and filtered variants were compared after excluding samples from patients #106, #113, #189, and #205, which had an abundance of noise. There was no significant difference in the amount of variant filtered out with FilterMutectCalls (Fig 8D, P = 0.4623) or variants with VAF <2.5% (Fig 8E, P = 0.9044). The detected variants were curated. Table 2 shows the pathogenic gene mutations, which were detected in 18 patients, with the initial genomic information obtained from nine of 11 patients without any karyotype abnormalities (eight normal, and three not available).

Mutations determining the disease subtype (*NPM1* and *CEBPA*) and germline mutations (*DDX41* and *RUNX1*) were particularly useful for a definitive diagnosis. Although target sequencing of the *CEBPA* gene is reportedly difficult [19], our assay was able to detect *CEBPA* mutations successfully in the smear samples. Moreover, prognostic factors, such as *TP53*, *FLT3*, and *ASXL1*, were also useful for determining indications for stem-cell transplantation. These findings demonstrated that archived smear samples can be used as templates for targeted DNA sequencing for molecular diagnosis.

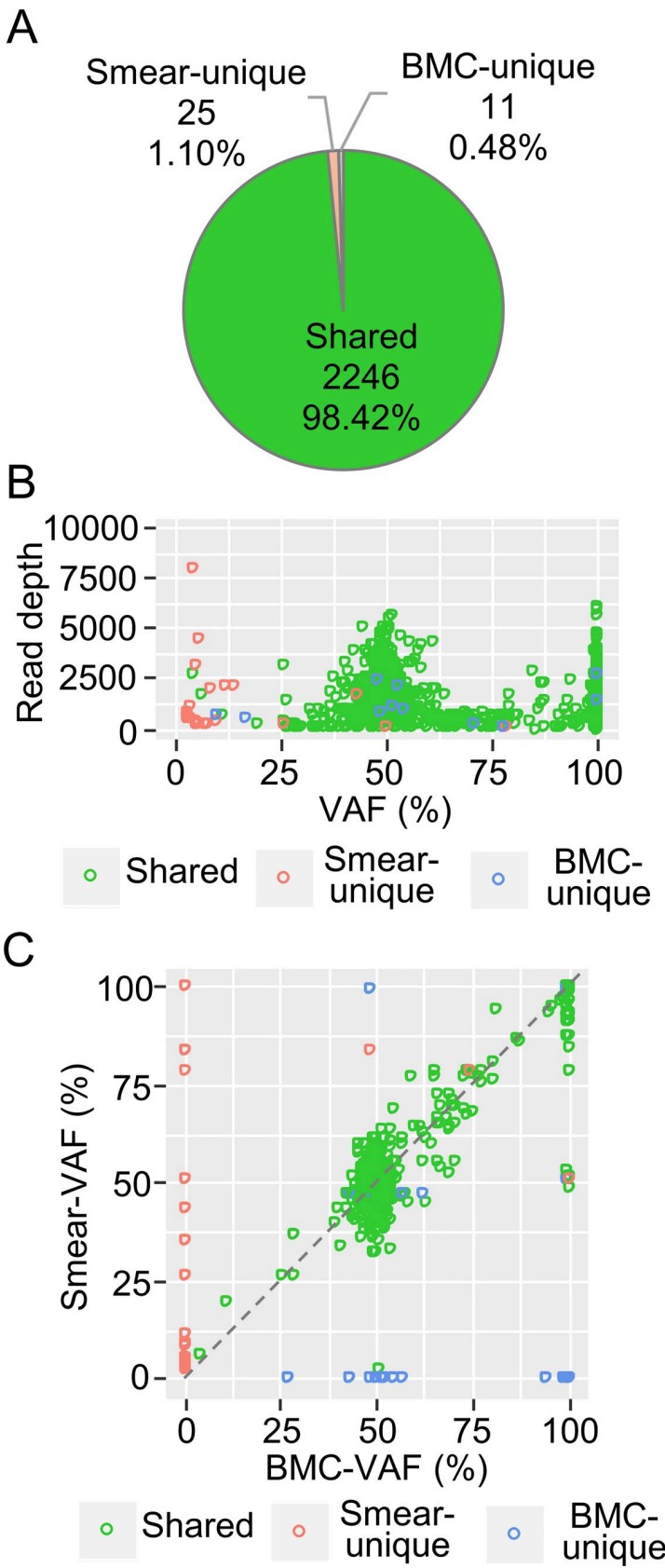

**Fig 6. Filtering effect on the variants detected using Mutect2 in Fig 4.** (A) Pie chart of the smear-unique, BMC-unique, and shared variants after filtering. (B) Distribution of the filtered variants. VAF, variant allele frequency. The smear-unique, BMC-unique, and shared variants were color-coded. (C) VAF plot of filtered variants in the BMC (X axis) and smear (Y axis) samples.

## Quality of RNA in smears and detection of fusion genes

RNA sequencing generally requires intact, high-quality RNA. However, targeted RNA sequencing can be performed if the desired fragments are amplified. In the present study, RNA was extracted from 15 smear samples and their fragmentation patterns were analyzed. Each RNA sample was sufficient for NGS analysis but displayed a very low fragment size (S2A Fig). The RIN value was also low independently of the duration from smear preparation to assessment, indicating that the RNA rapidly degraded with the start of smear preparation (Fig 9A). Nevertheless, reverse transcription was able to be performed even with the fragmented RNA, and libraries for targeted sequencing were fully synthesized (S2B Fig). Adequately-sized FASTQ files were generated through targeted RNA sequencing, and the obtained reads were able to be mapped to hg19. Among the detected fusions, highly expressed fusion genes identified using two detectors, JAFFA and STAR-Fusion, were considered as positive (Fig 9B). Fusion genes detected in five patients (#097 and #240 with *RUNX1-RUNX1T1*, #112 and #220 with *CBFB-MYH11*, and #238 with *ETV6-CHIC2*) were identical with their karyotypes, indicating that RNA from smears can be used to detect fusion genes via NGS (Table 3). Interestingly, unexpected fusion genes were detected through targeted RNA sequencing in two patients without translocation or inversion. The *KMT2A-MLLT10* fusion gene was identified in Patient #152 without the t(10;11) karyotype and confirmed by PCR (S3 Fig). Moreover, the

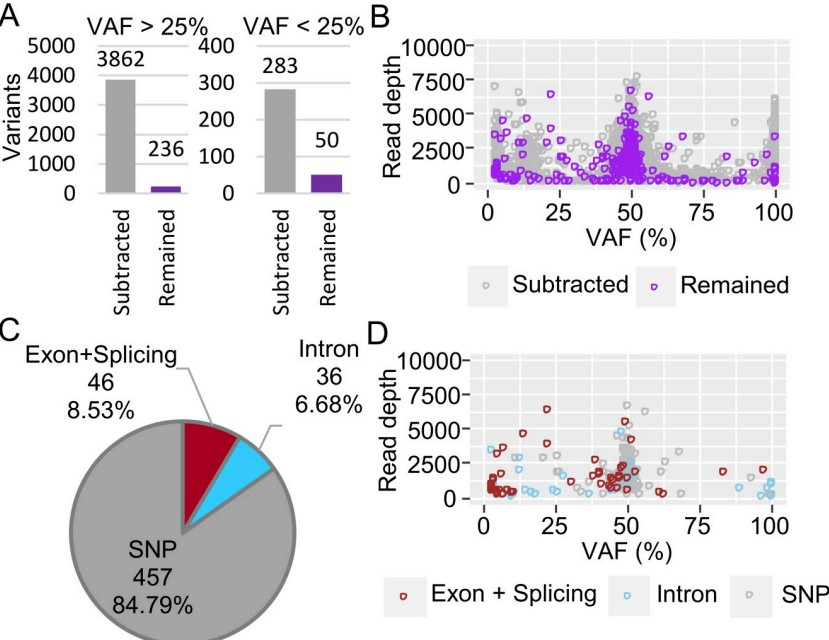

**Fig 7. Validation of PON filtration/depth filtration of the variants detected in 21 smear samples using Mutect2.** (A) Bar plot of the filtering effect. Variants with a VAF >25% are shown separately from those with VAF <25%. The subtracted variants are indicated in gray, and the remaining variants are indicated in purple. (B) Distribution of the subtracted and remaining variants. (C) Pie chart of the filtered variants. Known inherited germline variants (SNP), variants detected in exons and splice sites (exon+splice), and variants detected in introns (intron) are shown. (D) Distribution of the filtered variants.

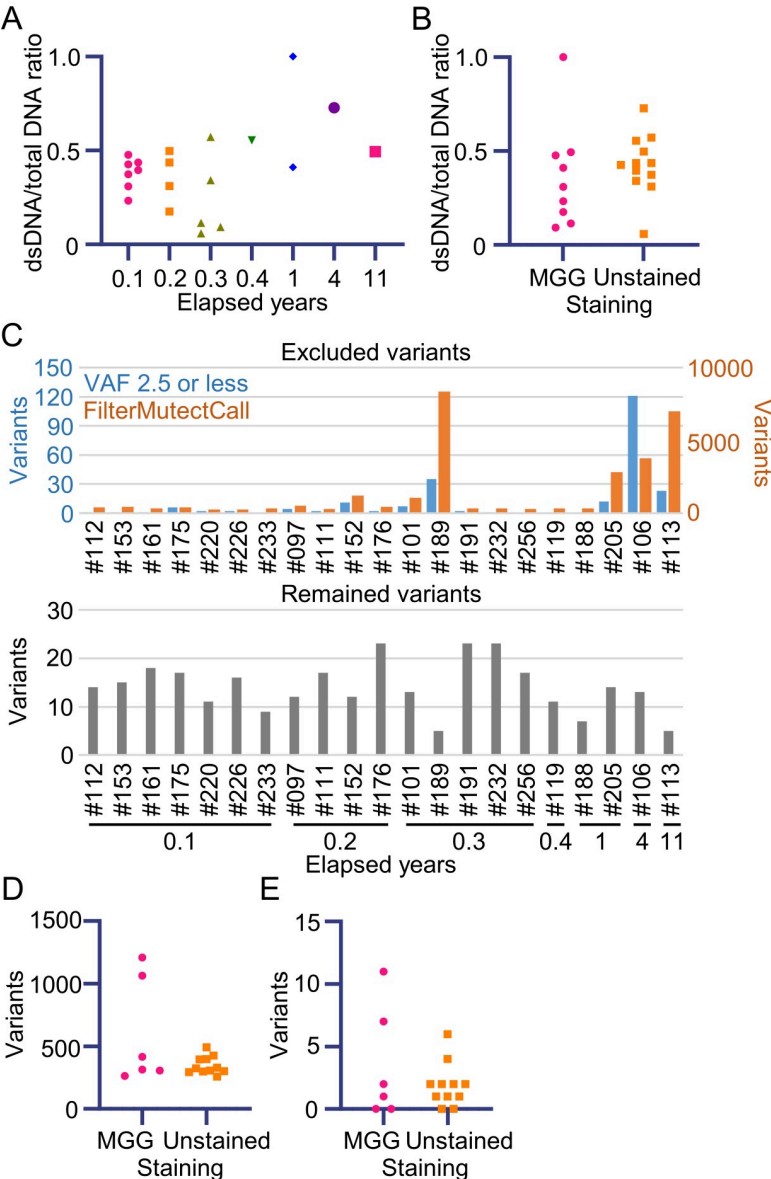

**Fig 8. Effect of duration and staining on DNA quality and variant calling in 21 smear samples.** (A) The dsDNA/total DNA ratio was plotted chronologically starting from smear preparation. (B) The dsDNA/total DNA ratio was compared between MGG-stained and unstained smear samples. (C) Bar plot of the amount of filtered and remaining variants. Filtered variants are shown separately by the filtering methods used (FilterMutectCall: orange; VAF 2.5 or less: blue) in the upper panel. Variants after filtering are shown in the lower panel. The quantity of variants removed by FilterMutectCalls (D) and the quantity removed at VAF 2.5 or a lower level of detection (E) were compared between the MGG-stained and unstained smear samples.

*NUP214-ABL* fusion gene, derived from t(9;9)(q34;q34) and difficult to detect by karyotypic analysis, was identified in Patient #231 and also confirmed by PCR (S4 Fig). These results underscored the utility of smear samples for diagnostic targeted RNA sequencing.

## Conclusions

The present results indicated that both DNA and RNA from smear samples can be used as templates for targeted NGS independently of the duration of preservation and staining. The

**Table 2. List of patients, sample status, and pathogenic genes detected in the DNA/RNA target sequencing data of the smear samples.**

| Patient | Clinical diagnosis | staining | Elapsed years | Mutation | | | |
|---|---|---|---|---|---|---|---|
| | | | | Gene | Nucleic acid | Amino acid | VAF (%) |
| #097 | AML | no | 0.2 | - | - | - | - |
| #112 | AML | no | 0.1 | *NRAS* | c.181C>A | p.Gln61Lys | 40.21 |
| #220 | t-AML | no | 0.1 | *NRAS* | c.182A>G | p.Gln61Arg | 46.84 |
| #226 | AML | MGG | 0.1 | *CEBPA* | c.912_913insTTG | p.Lys304_Gln305insLeu | 88.66 |
| | | | | *FLT3* | c.1747_1794dup | ITD16aa | 14.89 |
| #152 | AML-MRC | MGG | 0.2 | - | - | - | - |
| #111 | AML | no | 0.2 | *FLT3* | c.1836_1837insdup* | ITD17aa | 12.53 |
| #161 | AML | no | 0.1 | *IDH1* | c.395G>A | p.Arg132His | 44.53 |
| #175 | AML | no | 0.1 | *TP53* | c.488A>G | p.Tyr163Cys | 39.92 |
| #176 | AML | no | 0.2 | *ASXL1* | c.1605dupT | p.Pro536SerfsTer8 | 21.31 |
| | | | | *FLT3* | c.2039C>T | p.Ala680Val | 10.80 |
| #232 | AML | no | 0.3 | *SRSF2* | c.284C>A | p.Pro95His | 51.47 |
| | | | | *NPM1* | c.860_863dupTCTG | p.Trp288CysfsTer12 | 46.70 |
| #256 | AML | no | 0.3 | *CEBPA* | c.917_934delGCAACGTGGAGACGCAGC | p.Arg306_Gln311del | 49.78 |
| | | | | *CEBPA* | c.350dupG | p.Ala118ArgfsTer52 | 43.03 |
| | | | | *WT1* | c.1223T>A | p.Leu408Ter | 83.64 |
| | | | | *GATA2* | c.949A>G | p.Asn317Asp | 44.52 |
| #191 | AML | no | 0.3 | *NRAS* | c.34G>A | p.Gly12Ser | 37.97 |
| | | | | *TET2* | c.4144delC | p.His1382ThrfsTer66 | 46.88 |
| | | | | *TET2* | c.1842dupG | p.Leu615AlafsTer23 | 38.29 |
| | | | | *NPM1* | c.863_864insCATG | p.Trp288CysfsTer12 | 44.12 |
| | | | | *PTEN* | c.802-2A>T | Splicing | 5.51 |
| #205 | AML | MGG | 1 | *SRSF2* | c.284_307del | p.Pro95_Arg102del | 45.87 |
| | | | | *IDH2* | c.419G>A | p.Arg140Gln | 33.72 |
| | | | | *STAG2* | c.1810C>T | p.Arg604Ter | 27.27 |
| | | | | *STAG2* | c.2534-1G>A | Splicing | 6.69 |
| #153 | AML-MRC | MGG | 0.1 | *DDX41* | c.1496dupC | p.Ala500CysfsTer9 | 48.83 |
| | | | | *DDX41* | c.1574G>A | p.Arg525His | 12.16 |
| | | | | *SRSF2* | c.284C>G | p.Pro95Arg | 11.69 |
| #188 | aCML | MGG | 1 | *KRAS* | c.35G>T | p.Gly12Val | 37.97 |
| #101 | MDS | MGG | 0.3 | *U2AF1* | c.101C>T | p.Ser34Phe | 31.77 |
| #113 | MDS | MGG | 11 | *ATM* | c.3078delG | p.Trp1026CysfsTer3 | 6.09 |
| #233 | MDS | MGG | 0.1 | *TP53* | c.659A>G | p.Tyr220Cys | 18.65 |
| | | | | *TP53* | c.586C>T | p.Arg196Ter | 15.30 |
| #119 | MDS | no | 0.4 | *RUNX1* | c.417C>A | p.Asn139Lys | 48.05 |
| | | | | *RUNX1* | c.610C>T | p.Arg204Ter | 29.65 |
| | | | | *EZH2* | c.458A>G | p.Tyr153Cys | 40.73 |
| #189 | MDS | MGG | 0.3 | *TP53* | c.817C>T | p.Arg273Cys | 17.34 |
| | | | | *ASXL1* | c.2350delG | p.Asp784MetfsTer34 | 3.04 |
| #106 | MDS | no | 4 | - | - | - | - |

AML, acute myeloid leukemia; t, therapy-related; AML-MRC, AML with myelodysplasia-related changes; aCML, atypical chronic myeloid leukemia; MDS, myelodysplastic syndromes; MGG, May-Grünwald Giemsa; NA, not available.

*: c.1836_183 7insCGGC1788_1836dup.

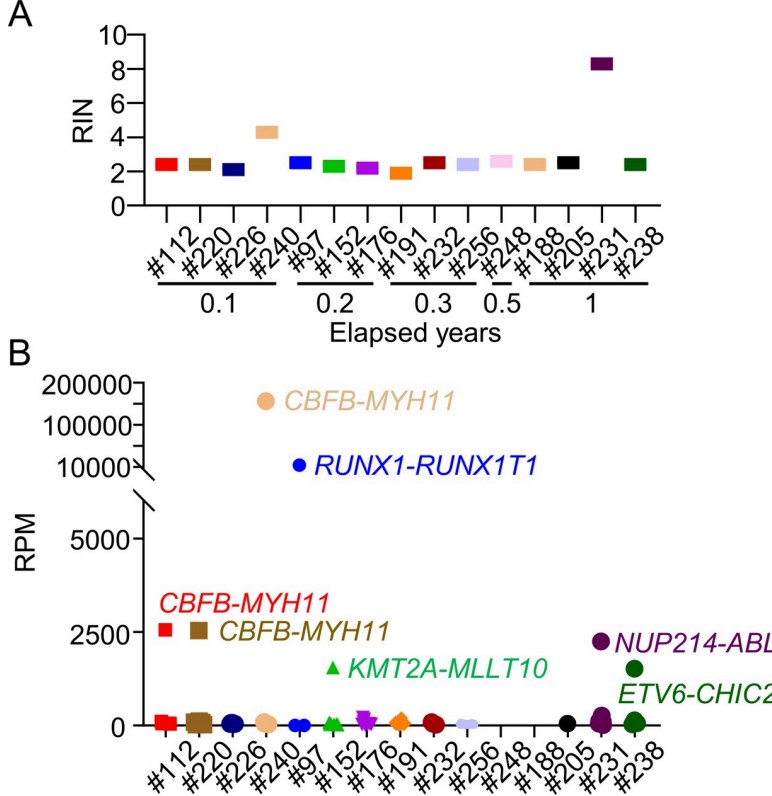

**Fig 9. Quality of smear-derived RNA and expression of fusion genes detected by targeted RNA sequencing.** (A) RIN value of each sample and elapsed years. (B) Reads per million mapped reads (RPM) of each sample were plotted. Highly expressed fusion genes are shown.

**Table 3. List of patients, karyotypes, and fusion genes detected in the RNA target sequencing data of the smear samples.**

| Patient | Clinical diagnosis | Karyotype | Mapped reads | Detected fusion gene | RPM |
|---|---|---|---|---|---|
| #097 | AML | 46,XY,t(8;21)(q22;q22)[17]/46,XY[3] | 33369 | *RUNX1-RUNX1T1* | 13828 |
| #112 | AML | 46,XX,inv(16)(p13.1q22)[20] | 35387 | *CBFB-MYH11* | 2560 |
| #152 | AML-MRC | 46,XX[20] | 48535 | *KMT2A-MLLT10* | 1551 |
| #176 | AML | 46,XY[20] | 44467 | - | - |
| #188 | aCML | 47,XY,+6[20] | 20447 | - | - |
| #191 | AML | 46,XX[20] | 84383 | - | - |
| #205 | AML | 46,XY[20] | 27515 | - | - |
| #220 | t-AML | 46,XX,inv(16)(p13.1q22)[20] | 48261 | *CBFB-MYH11* | 2546 |
| #226 | AML | 46,XX,i(7)(p10),-9,-9,+mar1,+mar2[20] | 31830 | - | - |
| #231 | MPAL | 46,XY,add(17)(p11.2)[12]/46,XY,del(17)(p?)[6]/46,XY[2] | 167249 | *NUP214-ABL* | 2255 |
| #232 | AML | 46,XY,del(11)(p?)[1]/46,XY[19] | 50825 | - | - |
| #238 | AML-MRC | 46,XY,t(4;12)(q12;p13)[14]/46,XY[6] | 30749 | *ETV6-CHIC2* | 1524 |
| #240 | AML | 46,XY,t(8;21)(q22;q22.1)[3]/46,idem,-Y[14]/46,idem,del(9)(q?)[2]/46,XY[1] | 56151 | *RUNX1-RUNX1T1* | 166556 |
| #248 | AML | 46,XX,+8[2]/46,XX[18] | 13875 | - | - |
| #256 | AML | 47,XY,+10[3]/46,XY[17] | 35635 | - | - |

AML, acute myeloid leukemia; t, therapy-related; AML-MRC, AML with myelodysplasia-related changes; aCML, atypical chronic myeloid leukemia; MPAL, mixed phenotype acute leukemia, t-AML, therapy-related acute myeloid leukemia.

variants detected in smear-derived samples were the same as those in BMC samples. Thus, pathogenic gene mutations and fusion genes can be detected from smear samples and can be especially useful for patients without karyotype abnormalities. Despite the generally inferior quality of their DNA and RNA, smear samples are useful for clinical molecular diagnosis as long as adequate noise-reduction methods are applied.

## Supporting information

**S1 Table. Target genes for targeted sequencing.**
(DOC)

**S1 Fig. Fragment analysis of synthesized libraries.** The fragment size (X axis) and fluorescent unit (Y axis) of synthesized libraries using smear-derived DNA (A) and bone marrow cell (BMC)-derived DNA (B) are shown. Yellow-highlighted regions indicate the predicted library size.
(TIF)

**S2 Fig. Fragment analysis of RNA extracted from smear samples and the synthesized libraries.** The fragment size (X axis) and fluorescent units (Y axis) of RNA (A) and the synthesized libraries (B) are shown. Yellow-highlighted regions indicate the predicted library size.
(TIF)

**S3 Fig. Detection of *KMT2A-MLLT10* fusion.** (A) The fusion sequence detected by targeted RNA sequencing is shown. Arrows indicate the primers for amplifying the target region. (B) A fusion gene confirmed by RT-PCR is shown. The following parameters were used with the PrimeSTAR GXL DNA Polymerase (TAKARA): 98˚C for 3 min, followed by 35 cycles at 98˚C for 10 s, 70˚C for 15 s, and 68˚C for 30 s. The sample from Patient#176 was used as a negative control.
(TIF)

**S4 Fig. Detection of *NUP214-ABL1* fusion.** (A) The fusion sequence detected by targeted RNA sequencing is shown. Arrows indicate the primers used to amplify the target region. (B) A fusion gene confirmed by RT-PCR is shown. The following parameters were used with the PrimeSTAR GXL DNA Polymerase (TAKARA): 98˚C for 3 min, followed by 35 cycles at 98˚C for 10 s, 75˚C for 15 s, and 68˚C for 30 s. The sample from Patient#176 was used as a negative control.
(TIF)

**S1 Raw images.**
(PDF)

## Author Contributions

**Conceptualization:** Daichi Sadato, Chizuko Hirama, Yuka Harada.

**Data curation:** Daichi Sadato, Yuka Harada.

**Funding acquisition:** Yuka Harada.

**Investigation:** Chizuko Hirama, Ai Kaiho-Soma, Ayaka Yamaguchi, Hiroko Kogure, Sonomi Takakuwa, Mina Ogawa.

**Methodology:** Daichi Sadato, Chizuko Hirama, Ai Kaiho-Soma.

**Resources:** Noriko Doki, Kazuteru Ohashi.

**Supervision:** Hironori Harada, Keisuke Oboki, Yuka Harada.

**Writing – original draft:** Daichi Sadato, Yuka Harada.

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
