## [Decision Letter · Decision Letter 0]

18 May 2021

PONE-D-21-08886

Archival bone marrow smears are useful in targeted next-generation sequencing for diagnosing myeloid neoplasms

PLOS ONE

Dear Dr. Harada,

Thank you for submitting your manuscript to PLOS ONE. After careful consideration, we feel that it has merit but does not fully meet PLOS ONE’s publication criteria as it currently stands. Therefore, we invite you to submit a revised version of the manuscript that addresses all the points raised during the review process by both Reviewers. Also, please improve the quality of the figures.

We look forward to receiving your revised manuscript.

Kind regards,

Francesco Bertolini, MD, PhD

Academic Editor

PLOS ONE

Journal Requirements:

2. Thank you for including your ethics statement: "The present study was approved by the institutional review board, and all the patients provided written informed consent."

3. Please include your tables as part of your main manuscript and remove the individual files. Please note that supplementary tables should remain uploaded as separate "supporting information" files

6. Please amend either the abstract on the online submission form (via Edit Submission) or the abstract in the manuscript so that they are identical.

Reviewers' comments:

Reviewer's Responses to Questions

**Comments to the Author**

1. Is the manuscript technically sound, and do the data support the conclusions?

Reviewer #1: Partly

Reviewer #2: Yes

2. Has the statistical analysis been performed appropriately and rigorously? 

Reviewer #1: No

Reviewer #2: I Don't Know

3. Have the authors made all data underlying the findings in their manuscript fully available?

Reviewer #1: Yes

Reviewer #2: Yes

4. Is the manuscript presented in an intelligible fashion and written in standard English?

Reviewer #1: Yes

Reviewer #2: Yes

5. Review Comments to the Author

Reviewer #1: Sadato et al reported very interesting result evaluating bone marrow slide as a potential source of DNA and RNA extraction for next-generation sequencing. This data is worthy of being shared with the community, but I am not sure it fits into the scope of PLOS ONE for publication. The novelty is relatively low-intermediate. The followings are my comment on the paper.

One of the critiques is consideration of time from marrow slide smear to the procurement for DNA/RNA extraction. As we are aware, DNA/RNA is degraded quickly over time. The authors presented elapsed time in Tables 1 & 2. But they have not analyzed the impact of elapsed time on the quality of DNA/RNA or the result of variant calling in DNAseq or fusion gene detection in RNAseq. This analysis is to be performed. I am not convinced that they have taken this part into consideration during their analysis of the data.

Also, the result comparing unstained smear slide vs. MGG stained slide is lacking. This information was presented in part, but I do not think it was analyzed systematically and compared between the two sources.

Also, given that slide storage is critical for the prevention of decay and degradation of genetic material, it would be critical to describe your method of slide storage in the paper.

The quality of Figures was very low.

Reviewer #2: Dr. Daichi Sadato and co-workers describe the use of bone marrow smears as a source of DNA and RNA for next generation sequencing analysis. Despite the inferior quality of the smear-derived material compared to nucleic acid obtained from paired fresh samples, the genetic analyses could be performed with success.

Interestingly, also extracted RNA provided useful information on known and unknown translocation. This is a nice piece of information, especially for retrospective studies or for patients with low material availability at diagnosis.

Major point:

-The HaplotypeCaller tool (for germline or large clone variant) found low percentage of smear and BMC-unique variants testifying a good correlation between variants found in fresh and smear derived material. Conversely, the Mutect2 tool (for somatic mutation) found much more variants (two-thirds of total) as smear-unique variants.

DNA single nucleotide variants (SNV) are per se neutral and stochastic events, therefore author should try to better explain the difference between the results of smear-unique mutations between the two bioinformatic tools.

6. PLOS authors have the option to publish the peer review history of their article (what does this mean?). If published, this will include your full peer review and any attached files.

Reviewer #1: No

Reviewer #2: No

---

## [Author Response · Author response to Decision Letter 0]

16 Jun 2021

Reviewer #1: Sadato et al reported very interesting result evaluating bone marrow slide as a potential source of DNA and RNA extraction for next-generation sequencing. This data is worthy of being shared with the community, but I am not sure it fits into the scope of PLOS ONE for publication. The novelty is relatively low-intermediate. The followings are my comment on the paper.

Thank you for your insightful comments, which have helped us significantly improve the paper. We added some new data and figure to address your questions below. 

One of the critiques is consideration of time from marrow slide smear to the procurement for DNA/RNA extraction. As we are aware, DNA/RNA is degraded quickly over time. The authors presented elapsed time in Tables 1 & 2. But they have not analyzed the impact of elapsed time on the quality of DNA/RNA or the result of variant calling in DNAseq or fusion gene detection in RNAseq. This analysis is to be performed. I am not convinced that they have taken this part into consideration during their analysis of the data.

Also, the result comparing unstained smear slide vs. MGG stained slide is lacking. This information was presented in part, but I do not think it was analyzed systematically and compared between the two sources.

We appreciate your observations and agree. We assessed the quality of all the DNA and RNA samples used in this study and analyzed the effect of the duration of preservation and staining. We added the findings in a new figure and described them in revised manuscript as follows.

・About the DNA

As a result of measuring and comparing the dsDNA/Total DNA ratio, we found that the quality of the DNA was not significantly affected by the length of preservation or by staining. However, for variant calling, samples with longer preservation times tended to have increasing quantities of noise that were able to be excluded by filtering. These results suggested that DNA could be extracted from various smears and that variant analysis could be done as long as allowances are made for the increased noise. We newly created Fig. 8 to show these results and added a discussion of this to Results and Discussion section (page 13, line 218 to page 14, line 229), and Conclusion section (page 20, line 296).

・About the RNA

We performed a new assessment of RNA quality using the RNA integrity number (RIN) with TapeStation (Agilent Technologies). The results are shown in Fig. 9A (newly added). The RIN values were generally very low, suggesting that RNA rapidly degraded as soon as the preparation of the smears but that some level of RNA quality, however low, could be maintained during storage. And as a result, the targeted RNA sequencing was successful. We added a discussion about the methods we used to the Materials and Methods section (page 5, line 78-80) and included the results in the Results and Discussion section (page 17, line 267-269) of the revised manuscript.

Also, given that slide storage is critical for the prevention of decay and degradation of genetic material, it would be critical to describe your method of slide storage in the paper.

As pointed out, the conditions of slide storage are important information. The smear slides used in this study were stored in a clinical laboratory (at room temperature in a dark place) with no other special care. We added this information to Materials and Methods (page 5, line 72), Results and Discussion (page 13, line 207-208). 

The quality of Figures was very low.

We apologize for the quality of the figures. We replaced them all with high resolution images.

Reviewer #2: Dr. Daichi Sadato and co-workers describe the use of bone marrow smears as a source of DNA and RNA for next generation sequencing analysis. Despite the inferior quality of the smear-derived material compared to nucleic acid obtained from paired fresh samples, the genetic analyses could be performed with success.

Interestingly, also extracted RNA provided useful information on known and unknown translocation. This is a nice piece of information, especially for retrospective studies or for patients with low material availability at diagnosis.

Thank you for your careful reading and helpful comments on our study. We analyzed and re-checked the data to confirm our results.

Major point:

-The HaplotypeCaller tool (for germline or large clone variant) found low percentage of smear and BMC-unique variants testifying a good correlation between variants found in fresh and smear derived material. Conversely, the Mutect2 tool (for somatic mutation) found much more variants (two-thirds of total) as smear-unique variants. N 

DNA single nucleotide variants (SNV) are per se neutral and stochastic events, therefore author should try to better explain the difference between the results of smear-unique mutations between the two bioinformatic tools.

We appreciate the reviewer's observation on this point. The difference in the quantity of smear-unique mutations is due to different levels of sensitivity of the assessment tools to low VAF variants. In principle, Mutect2 can detect more low-VAF variants and thus more smear-sample-derived noise than HaplotypeCaller. We added this observation to Results and Discussion (page 13, line 208-213).

---

## [Decision Letter · Decision Letter 1]

13 Jul 2021

Archival bone marrow smears are useful in targeted next-generation sequencing for diagnosing myeloid neoplasms

PONE-D-21-08886R1

Dear Dr. Harada,

We’re pleased to inform you that your manuscript has been judged scientifically suitable for publication and will be formally accepted for publication once it meets all outstanding technical requirements.

Kind regards,

Francesco Bertolini, MD, PhD

Academic Editor

PLOS ONE

Additional Editor Comments (optional):

Reviewers' comments:

Reviewer's Responses to Questions

**Comments to the Author**

1. If the authors have adequately addressed your comments raised in a previous round of review and you feel that this manuscript is now acceptable for publication, you may indicate that here to bypass the “Comments to the Author” section, enter your conflict of interest statement in the “Confidential to Editor” section, and submit your "Accept" recommendation.

Reviewer #1: All comments have been addressed

Reviewer #2: All comments have been addressed

2. Is the manuscript technically sound, and do the data support the conclusions?

Reviewer #1: Yes

Reviewer #2: Partly

3. Has the statistical analysis been performed appropriately and rigorously? 

Reviewer #1: Yes

Reviewer #2: I Don't Know

4. Have the authors made all data underlying the findings in their manuscript fully available?

Reviewer #1: Yes

Reviewer #2: Yes

5. Is the manuscript presented in an intelligible fashion and written in standard English?

Reviewer #1: Yes

Reviewer #2: Yes

6. Review Comments to the Author

Reviewer #1: All the queries are answered soundly. Thus this work is acceptable for publication. It is not a major production, but requires to be shared in the community.

Reviewer #2: Your reply only parially explained tthe difference in findings of the two sistems. Low levels, expecially of poi t mitation, also can only be sequence errors.

7. PLOS authors have the option to publish the peer review history of their article (what does this mean?). If published, this will include your full peer review and any attached files.

Reviewer #1: No

Reviewer #2: No

---

## [Editor Report · Acceptance letter]

15 Jul 2021

PONE-D-21-08886R1 

Archival bone marrow smears are useful in targeted next-generation sequencing for diagnosing myeloid neoplasms 

Dear Dr. Harada:

I'm pleased to inform you that your manuscript has been deemed suitable for publication in PLOS ONE. Congratulations! Your manuscript is now with our production department. 

Kind regards, 

on behalf of

Dr. Francesco Bertolini 

Academic Editor

PLOS ONE